# Effect of a 14-Day Period of Heat Acclimation on Horses Using Heated Indoor Arenas in Preparation for Tokyo Olympic Games

**DOI:** 10.3390/ani14040546

**Published:** 2024-02-06

**Authors:** Carolien Munsters, Esther Siegers, Marianne Sloet van Oldruitenborgh-Oosterbaan

**Affiliations:** 1Department of Clinical Sciences, Faculty of Veterinary Medicine, Utrecht University, Yalelaan 114, 3584 CM Utrecht, The Netherlandsm.sloet@uu.nl (M.S.v.O.-O.); 2Equine Integration B.V., Groenstraat 2c, 5528 Hoogeloon, The Netherlands

**Keywords:** equine, welfare, heat, acclimation, exercise

## Abstract

**Simple Summary:**

In order to help horses perform better and stay healthy in hot climates during equestrian competitions, it is recommended to get them acclimated to heat. The effects of training in a heated indoor arena on four Olympic horses (13.3 ± 2.2 years; three eventers, one para-dressage horse) were studied. The horses were trained in a heated indoor arena (32 °C with 50–60% humidity) for 14 days following their normal training routine in preparation for the Tokyo Olympic Games. Before and after this acclimation period, standardized exercise tests were performed to measure various factors like heart rate, lactate concentrations, rectal temperature, sweat loss, and sweat composition. After the 14-day training period in the heat, the horses showed positive changes. Their body temperature and heart rate decreased, and it took longer before they reached their highest body temperature, meaning they were better able to dissipate heat and experienced reduced stress. In conclusion, training for 14 days in a heated indoor arena helped reduce the thermal strain on elite sport horses, making it easier for them to compete in hot weather. This study showed that top-level horses can acclimate to heat while following their regular training routines, which is important for participating in competitions like the Olympics.

**Abstract:**

To optimise the performance and welfare of horses during equestrian competitions in hot climates, it is advised to acclimate them to the heat. The effects of training in a heated indoor arena were studied. Four Olympic horses (13.3 ± 2.2 years; three eventers, one para-dressage horse) were trained for 14 consecutive days in a heated indoor arena (32 ± 1 °C; 50–60% humidity) following their normal training schedule in preparation for the Tokyo Olympic games. Standardised exercise tests (SETs) were performed on Day 1 and Day 14, measuring heart rate (HR; bpm), plasma lactate concentration (LA; mmol/L), deep rectal temperature (T_rec_; °C), sweat loss (SL; L), and sweat composition (K^+^, Cl^−^ and Na^+^ concentration). The data were analysed using linear mixed models. The T_rec_ and HR were significantly decreased after acclimation (estimate: −0.106, 95% CI −0.134, −0.078; estimate: −4.067, 95% CI −7.535, −0.598, respectively). Furthermore, for all the horses, the time taken to reach their peak T_rec_ and heat storage increased, while their LA concentrations decreased. The SL, Cl^−^, and Na^+^ concentrations decreased in three out of the four horses. Conclusions: Fourteen days of normal training in a heated indoor arena resulted in a reduction in cardiovascular and thermal strain. This is advantageous because it shows that elite sport horses can be acclimated while training as usual for a championship.

## 1. Introduction

With regards to the World Equestrian Games in Stockholm (1990) and the Olympic Games in Barcelona (1992), the media paid significant attention to 3-day eventing horses exhibiting signs of heat stress [1]. Because of the veterinary health concerns related to heat stress, multiple scientific studies were performed on how to safeguard equestrian competitions in challenging environmental conditions [2,3,4,5,6,7,8]. Over time, the public’s perception of the use of horses for sport and entertainment has evolved. There is an increasing public scrutiny of the potential effect of current equine activities and practices on equine welfare, questioning the equestrians’ social license to operate (SLO) [9,10]. Adding to this, as a result of the evidence of a changing climate, an increasing number of horses will face challenging thermal conditions when being exercised or housed. As shown by Takahashi et al. [11] and Trigg et al. [12], exercising in hot conditions increases the risk of heat-related illnesses in horses. Therefore, it is crucial to gain a better understanding of methods that can be employed to help horses alleviate thermal strain when performing in thermally challenging climates.

It is known that exercising in hot and humid conditions (at an ambient temperature of around 30–34 °C and a relative humidity (RH) of around 75–85%) reduces the exercise capacity of horses compared to a more thermoneutral condition [7,13]. High environmental temperatures increase heat strain in (equine) athletes. Additionally, high humidity may negatively affect performance further, and it increases the risk of exhaustion and heat stroke [2,7,14].

The Tokyo Olympic and Paralympic Games in 2021 were expected to be held in the hottest conditions compared to previous Olympic games. Under these conditions, human and equine athletes face exceptionally challenging situations that may impede their optimal performance and increase the risk of heat-related illnesses. It is widely acknowledged that heat acclimation (in a climate chamber) or acclimatisation (in a natural environment) induces physiological adaptations that mitigate thermal strain through improved thermoregulatory capabilities [2,15,16]. Therefore, it was already suggested before the Olympic Games in Atlanta, 1996, to acclimate horses for at least two weeks in hot conditions. Alternatively, acclimatisation could be facilitated by relocating the horses to the competition region or to a location with environmental conditions comparable to those at the competition site two weeks prior to the event.

Regarding horses, only two studies have been performed on heat acclimation. Both studies demonstrated physiological adaptations indicative of improved heat tolerance. In a study by Marlin et al. [17], five horses were trained on a treadmill under hot and humid conditions (a WBGT; Wet-Bulb Globe Temperature, of 28 °C) for 15 days, exercising 80 min/day and resting 20 min/day in these circumstances. In a study by Geor et al. [15], six horses were trained for 1 h for 21 days on a treadmill under hot and humid conditions (a WBGT of 32 °C) and exposed, in total, for 4 h daily to these conditions. These findings have been disseminated across multiple papers, each going into distinct facets of heat acclimation in horses [15,18,19].

In another study by Marlin et al. [20,21], six horses underwent a 16-day period of acclimatisation by transporting European eventing horses to the event site prior to the event at a 26 °C WBGT. These horses were trained for around 70 min/day, with heart rates below 120 bpm on most days. Compared to the baseline period in Europe (WBGT around 22 °C), the respiratory rate increased throughout the period, but the rectal temperatures (T_rec_), heart rate (HR), and plasma volume at rest were unchanged. This shows that horses that have not undergone prior acclimatisation can effectively adapt to these stressors and, with proper care, stay physically fit and healthy with a reduced risk of heat-related illnesses. The relocation of horses to train them for several weeks and let them adapt to hotter climates can be useful, but it involves significant logistics and costs [22]. The acclimation studies were performed with horses exercising in a heated treadmill room. This may not be favourable for sport horses, as most of these horses are not used to treadmill exercise, and this might interfere with their normal training regime. Especially for high-level competitions, it is not beneficial to change training regimes or induce new training types before a competition. Other practical solutions need to be found to assist horses in acclimating with the least interference to their normal training habitats.

Therefore, the aim of this study was to investigate the effects of a 14-day period of heat acclimation by training in a heated indoor arena on four elite sport horses in preparation for the 2021 Olympic Games in Tokyo.

## 2. Materials and Methods

### 2.1. Horses

Data were collected from four privately owned Olympic horses (three eventing horses, one para-dressage horse (grade IV)). These horses were, on average, 13.3 ± 2.2 years old, weighing 563 ± 89 kg (range 458–675 kg) with a wither height of 1.70 ± 0.07 m (range 1.61–1.78 m). Horses were selected to participate at the 2021 Olympic Games in Tokyo under the rules of the Federation Equestrian International (FEI) and underwent full preparation for this competition. Horses had to be qualified and selected to perform at the Olympic Games in Tokyo, and horse owners and riders had to be willing to participate in the study. During the study, horses were ridden by their usual rider, housed in individual stables, and provided with daily turnout. They had access to water ad libitum and were fed an individual diet. The Animal Ethics Committee of Utrecht University approved all procedures (reference number: 5204-1-3; approval date: 14 June 2019). The owner’s written informed consent was obtained for each horse, and rider consent was similarly secured.

### 2.2. Study Design

The study consisted of a 14-day acclimation period with a discipline-specific exercise test performed on two occasions: on the first day of acclimation (Day 1) and on the last day of acclimation (Day 14). The acclimation period took place three weeks before the actual competition date at the Olympic Games in Tokyo.

Two heated indoor arenas were used for the study, as the three eventing horses participated in the Olympic games in July 2021 and the para-dressage horse participated in the Paralympic games in August 2021. The three eventing horses were all stabled together, using their regular indoor arena (60 × 30 m). In addition, a second indoor arena (40 × 20 m) was prepared to accommodate the daily training of the para-dressage horse. Both indoor arenas were equipped with powerful heating systems designed for large spaces to maintain a temperature of 32 ± 1 °C during training times. Furthermore, the humidity was increased to 50–60% by large humidifiers. Further adjustments to temperature or humidity were unfeasible due to the extensive size of the arenas. The heaters and humidifiers were activated one hour prior to the training session, ensuring ample time for the climate to reach its desired conditions. The heat and humidity were intended to simulate conditions during the Tokyo Olympic Games as much as possible [16].

#### 2.2.1. Acclimation Period

Before the acclimation period, the horses were trained mainly outdoors in a moderate European (Dutch) climate. The ambient temperature and relative humidity (RH) two weeks before acclimation were, on average, 18.7 ± 3.6 °C and 71.9 ± 10.3% RH for the eventing horses and 18.5 ± 1.1 °C and 79.4 ± 7.4% RH for the para-dressage horse.

During the acclimation period, all horses were trained for 60 min per day in the heated indoor arena (32 ± 1 °C, 50–60% RH). During the acclimation period, the horses continued with their normal daily training routine as much as possible. The eventing horses had to perform two outdoor training sessions during this period, which could not be performed in an indoor arena due to the required higher speeds or hills. After these training sessions, the horses performed one hour of hacking work in the heated indoor arena.

#### 2.2.2. Standardised Exercise Tests

All SETs consisted of three phases: A warm-up phase of 26 min at a walk, trot and canter (~5.3 m/s). Subsequently, a discipline-specific phase of 11 min followed. For the eventing horses, this phase included two bouts of 4.5 min at a canter (~7.2 m/s), with 2 min of walking in between bouts. The para-dressage horse performed 6 min of competition-specific exercises at a trot, followed by 5 min of competition-specific exercises at canter during the discipline-specific phase. All horses concluded with a 10 min recovery phase at a walk.

### 2.3. Measurements

During the standardised exercise tests (SETs) performed on Day 1 and Day 14 of the acclimation period, heart rate (HR; bpm) was measured and recorded (Polar V800^®^ Polar Electro Oy, Kempele, Finland). Blood samples were taken from a jugular vein 60–90 s after finishing the training-specific period of the SETs and again after 10 min of recovery for the measurement of plasma lactate concentrations (LA, mmol/L; Lactate Pro 2^®^ Arkray Inc., Kyoto, Japan). The bodyweight of the horses was measured before saddling and after finishing the SETs (horse scale: custom made: Breinler International; platform scale: SATEX 34SA-1 250, Weegtechniek Holland, Barneveld, The Netherlands). Faeces produced between the first and second weighing moments were collected and also weighed. The ambient temperature, relative humidity (RH; %), and Wet-Bulb Globe Temperature (WBGT; °C) were obtained at the start of each SET (ExTech HT30, ExTech, Pittsburgh, PA, USA). A rectal temperature sensor (B10014, MSR, Seuzach, Switzerland) was inserted to a depth of 25 cm into the rectum with a datalogger attached to the base of the tail, measuring rectal temperature (T_rec_; °C) continuously. The time (s) to peak T_rec_ was calculated. Heat storage (S; kJ/m^2^) was estimated [23] at the end of the most intense exercise bout (discipline-specific phase; S_ex_, and after 10 min of walking recovery; S_rec_). The change in heat storage (∆S) was calculated using Equation (1) [23]:S (∆S) = 3.48 × body mass (kg) × ∆T_rec_/body surface area (m^2^)(1)

Pre-exercise body mass (kg) was used. To calculate S_ex_, ∆T_rec_ was used, which reflects the change in temperature during the exercise period by subtracting the end-exercise temperature (after the last intense exercise bout) from the resting temperature. To calculate S_rec_, T_rec_ at the end of the recovery period was subtracted from the end-exercise temperature. The specific heat capacity of a horse is not known; therefore, the value for humans (3.48 kJ/kg/°C) was used. The body surface area (SA; m^2^) was calculated by using the formula SA = 1.09 + 0.008 × body mass [24]. In the present study, it was not possible to let the horses stand in the heated indoor arena for an additional recovery period of 60 or 120 min, as carried out in a study by Geor [23]. However, to obtain a comparable measure for S, the total heat storage till the end of all the exercises (the end of the recovery period of walking) was also calculated (S_tot_).

Sweat loss (SL) was calculated using Equation (2):SL = (*m*_horse-post_ − *m*_horse-pre_) + *m*_faeces_(2)

Here, *m* represents mass (kg). Sweat compensation was evaluated using the absorbent patch technique [24], employing two custom-made absorbent patches (each 25 cm²) affixed to the neck and gluteus using removable glue [25]. Sweat sodium, chloride, and potassium levels were measured via ion-selective electrodes (Cobas 8000, Roche Diagnostics, Indianapolis, IN, USA) after centrifuging the patches in sealed plastic tubes at 1500× *g* for 5 min. The background ion concentrations of the patches were analysed, and necessary corrections were applied accordingly (sodium: −7.9 mmol/L and chloride: −6.6 mmol/L). In addition, plasma serum amyloid A (SAA; µg/mL; Stablelab^®^ EQ-1, Zoetis, Parsipanny, NJ, USA) was measured on Days 1, 2, and 14.

Given that this study involved animals preparing for an actual high-level competition, the capacity to delve into individual classic acclimatisation responses was restricted compared to studies conducted in laboratory settings with controlled environments and treadmills.

### 2.4. Data Analysis

Continuously recorded data (HR and T_rec_) were transformed to 1 min averages. To check whether the normality assumption was reasonable, normality probability plots of the residuals were made. If the normality assumption did not hold, the data were log-transformed to obtain a normal distribution. HR and T_rec_ were analysed using a linear mixed-effect model with horse as a random effect and time (minutes), SET, and discipline as fixed effects. This was called the starting model. Akaike’s Information Criterion (AIC) was used for model reduction. The model for which there was no further reduction possible was taken as the final model, and the remaining factors were considered important. The 95% likelihood confidence intervals (95% CI) were reported where there was an important effect.

## 3. Results

### 3.1. General

The ambient temperature, relative humidity, and WBGT of the heated indoor arena during SET 1 were, respectively, 31.9 ± 2.0 °C, 57.7 ± 9.2%, and 27.9 ± 0.9 °C, and during SET 2, they were 33.8 ± 0.5 °C, 48.4 ± 4.4%, and 27.6 ± 0.5 °C. There was no difference in the ambient temperature, RH, or WBGT between the heated indoor arenas. All the horses remained fit and healthy throughout the acclimation period and participated successfully at the Olympic Games three weeks later.

### 3.2. Heart Rate and Plasma Lactate Concentration

The time and SETs had important effects on HR. Overall, HR was lower during SET 2 compared to SET 1 (estimate: −4.067, 95% CI: −7.535, −0.598), as shown in Figure 1. After ten minutes of exercise, HR increased significantly compared to at the start of the SETs.

The mean LA concentration directly after the discipline-specific phase and after 10 min of recovery was 2.3 ± 0.5 and 1.1 ± 0.1 mmol/L, respectively, during SET 1 and 1.7 ± 0.4 and 1.1 ± 0.3 mmol/L during SET 2. The LA concentration after the discipline-specific phase decreased in all the horses during SET 2 compared to SET 1; see Figure 2a,b.

### 3.3. Rectal Temperature

T_rec_ was highly associated with time and the SETs. The T_rec_ of the horses was lower during SET 2 compared to SET 1 (estimate: −0.106, 95% CI: −0.134, −0.078), as shown in Figure 3. After 12 min of exercise, T_rec_ significantly increased till the end of the SETs compared to at the start of the SETs; see Figure 3. There was no effect of discipline on T_rec_. The time to reach the peak T_rec_ increased for all the horses during SET 2 compared to SET 1, whereas, on average, the peak T_rec_ was reached during SET 1 after 40.8 ± 9.0 and during SET 2 after 46.0 ± 7.9 min; see Figure 2c.

### 3.4. Heat Storage

Total heat storage (S_tot_), S_ex_, and S_rec_ during SET 1 and SET 2 are shown in Table 1. The total heat storage increased for all the horses from SET 1 to SET 2.

### 3.5. SL and Sweat Composition

The average body weight (BW) of the horses before SET 1 was 563.5 ± 89.1 kg, and before SET 2, it was 559 ± 87.2 kg. However, there were large individual variations (see Figure 2d). The BW decreased for two horses, increased for one horse (Horse 1), and stayed the same for Horse 3. The sweat loss (SL) after acclimation decreased in three out of the four horses during SET 2 compared to SET 1, except for Horse 2, which lost more fluid compared to SET 1; see Figure 2d. After SET 1, the mean SL was 9.4 ± 4.9 L, and after SET 2, it was 7.4 ± 3.8 L. The percentage of BW loss during SET 1 was 1.6 ± 0.6%, and during SET 2, it was 1.3 ± 0.7%, whereas Horse 2 exhibited an increase in terms of fluid loss (from 0.8 to 1.9 during SET 2). The reduction in fluid loss from SET 1 to SET 2 in Horses 1, 3, and 4 was 41.9 ± 19.0%.

The data from the sweat patches on the neck could not be used for the data analysis as two out of the four horses lost their patch in this location due to heavy sweating. Therefore, only the data from the sweat patch of the gluteus are presented. The mean electrolyte concentrations seemed to decrease after the acclimation period, as shown in Table 2. However, there were individual effects. All the horses had lower sweat chloride and sodium concentrations after acclimation except for Horse 2; see Figure 4.

### 3.6. SAA

The SAA concentration was 0.0 µg/mL for three out of the four horses during SET 1 (Day 1), and Horse 4 had an SAA concentration of 19.0 µg/mL during SET 1. On Day 2, three out of the four horses had an SAA concentration of 0.0 µg/mL, and Horse 2 had an SAA concentration of 1.0 µg/mL. During SET 2 (Day 14), all the horses had an SAA concentration of 0.0 µg/mL.

## 4. Discussion

This is the first study showing that an acclimation period of 14 days, training for 60 min per day in a heated indoor arena (32 ± 1 °C, 50–60% RH) can be used during real-life preparation for a high-level competition to reduce the thermal strain in horses. After the acclimation period, the horses showed decreased heart rate, rectal temperature, and LA concentration, an increase in the time taken to reach the peak rectal temperature, and an increase in heat storage. Other studies on acclimation in horses indicated that training horses in a hot and humid environment for a period of 15 to 21 days is effective in eliciting the required physiological responses to enhance heat tolerance. It is worth noting that in those studies, horses were also subjected to a hot and humid environment, both actively (exercise) and passively, with rest durations ranging from 20 min to 4 h [15,17,23]. The current study demonstrates that training horses for only 60 min in a hot and humid environment was sufficient to achieve comparable physiological adaptations.

In humans, heat acclimation induces adaptations which contribute to a reduced thermal strain during exercise at a specific workload, typically manifested by a lower core temperature, heart rate, and skin temperature, along with an increased whole-body sweat rate and improved thermal comfort [26,27]. The present study shows comparable improvements in horses after an acclimation period. However, their skin temperature, sweating rate, and plasma volume could not be obtained due to the practical limitations of working with elite horses during normal training circumstances. The reduction in T_rec_ observed in the current study after acclimation is consistent with results reported previously [15,17,28]. Notably, it was observed that after acclimation, T_rec_ was lower during exercise, but it was higher during the recovery period compared to pre-acclimation. Marlin et al. [17] showed similar findings. Furthermore, the temperature measured in the pulmonary artery (T_pa_) was significantly higher during the recovery period than during the exercise period.

In addition, in the present study and in the study by Marlin et al. [17], the horses displayed an increase in their total heat storage (S_tot_) after acclimation. This diverges from the results reported by Geor et al. [15,23]. In all these studies, the horses experienced a decrease in resting rectal temperatures (approximately 0.3 °C). In the present study and in the study by Marlin et al. [17], an increase in the exercise time to reach the peak T_rec_ was found, and the horses showed a more rapid increase in rectal temperature post-acclimation, implying greater heat storage. In addition, in the present study, two out of the four horses achieved a higher peak T_rec_ without displaying signs of fatigue (and with lower LA and HR values compared to SET 1), indicative of improved heat tolerance or improved heat transfer. Nevertheless, having greater heat storage after acclimation may be evidence that a horse is functioning near its limits of adaptation [17]. As the exercise intensity and ambient conditions varied significantly across the studies, potentially influencing the divergent outcomes observed, further investigation in the future is warranted to better understand the underlying causes and the practical implications of exercising in the heat.

The present study showed a decrease in HR after acclimation, which is consistent with equine [15,28] and human studies. A reduction in heart rate during consecutive SETs in a hot environment may signify a genuine heat-adaptive response. However, in horses, other studies show contradictory results; some studies in horses demonstrate a decrease in HR [15,28], and others, such as that by Marlin et al. [17], do not report a similar trend. However, these results cannot be directly compared as the exercise intensity and duration and the ambient conditions were not exactly the same. In addition, it is worth noting that in the present study, the use of large heaters and humidifiers in the indoor arena could have partly contributed to the lower post-acclimation HR, as some response to this equipment might have caused an increase in HR in the first SET, and the horses may have gradually become accustomed to the sounds and airflow generated by these machines during the 14 days of acclimation. On the other hand, in addition to the decrease in heart rate after acclimation, there was also a reduction in plasma lactate concentrations. Together, these changes in physiological responses seem to indicate a reduction in thermal strain after acclimation and not only habituation to the heating equipment. In the study by Marlin et al. [17], no reduction in LA concentration after acclimation was seen, but a reduction in oxygen consumption of ~8% was shown. In humans, it is suggested that acclimation improves muscle efficiency, with heat acclimation being associated with a reduction in aerobic metabolic rate as well as decreased muscle and blood lactate accumulation during exercise [27,29]. In this study, while oxygen consumption could not be directly measured, the observed decrease in lactate concentrations appears to be associated with a reduced workload on the heart and muscles, suggesting an improvement in performance.

In humans, it has been shown that individual characteristics are associated with the magnitude of heat acclimation adaptations (low, moderate, or high responders), and attempts are made to identify which factors may predispose individuals to either group [26]. Genetic components, fitness status, and body dimensions may be related to inter-individual differences in adaptive responses to heat acclimation [26,30]. In horses, only empirical evidence seems to indicate that horses also show individual responses to heat acclimation. For significant practical importance, equestrian and welfare organisations and horse trainers/owners might individualise strategies that reduce the thermal strain on horses. Some horses may even seem to be heat-intolerant [31].

Although it was not the goal of the present study to identify factors associated with individual differences in heat acclimation adaptations, clear individual differences were observed. While after acclimation, a reduction in HR, LA, and T_rec_ occurred and the time to peak T_rec_ and heat storage increased in all the horses, the sweat loss, chloride, and sodium concentrations decreased only in three out of the four horses. At the commencement of the acclimation period, Horse 2 exhibited the lowest SL among all the horses (3.8 L), while the remaining horses demonstrated higher sweat losses during SET 1 (ranging from 8.8 to 15.8 L). While SL decreased in the other horses over the acclimation period, Horse 2 showed an increase in SL (reaching 8.8 L during SET 2). In addition, the sweat composition changed in all the horses from SET 1 to SET 2. However, Horse 2 showed an increase in all the measured sweat ion concentrations, which seems to be in line with the increase in SL seen in this horse. It can be hypothesised that an acclimation period of more than 14 days would have been beneficial for this horse. Geor et al. [15] showed that a clear decrease in fluid loss appeared on Day 15 and Day 20 of acclimation, but this was not yet shown on Day 10. However, the rectal temperature of Horse 2 did decrease during the 14-day acclimation period, suggesting that the horse did show signs of heat acclimation adaptations but was capable of relying on a different strategy to reduce heat strain. This shows the importance of monitoring the individual responses of horses to a period of heat or acclimation.

Horse 1, Horse 3, and Horse 4 showed a reduction in sweat sodium and chloride concentrations after acclimation. In a study by McCutcheon et al. [32], the sweat sodium concentration was reduced on Day 14 and Day 21 after acclimation. They showed a decrease of 10% in the mean sodium concentration in sweat fluid for a given sweat rate. It is concluded that horses can respond to heat acclimation with a reduction in sweat ion losses, but individual differences might also occur.

Before acclimation, the magnitudes of fluid loss for the other three horses were comparable with the results found by Geor et al. [15] and Marlin et al. [17]. After acclimation, the reduction in fluid loss closely mirrored the percentages observed in the study by Marlin et al. [17], ranging from approximately 3.8% (pre-acclimation) to 2.1% (post-acclimation), resulting in a 45% decrease. In contrast, Geor’s study showed a reduction from 2.7% (pre-acclimation) to 2.2% (post-acclimation), yielding a 19% decrease. In the current study, the pre-acclimation fluid losses in the remaining three horses averaged 1.9 ± 0.4%, while the post-acclimation losses averaged 1.1 ± 0.6%, indicating an average reduction of 41.9%. The larger reduction in fluid loss might also partly explain why heat storage increased post-acclimation in the present study and in the study by Marlin et al. [17]. As suggested in earlier research, horses already have very high sweating rates. Therefore, it is questionable whether horses are able to raise this sweating rate even further in challenging conditions. The decrease in fluid loss seems to be primarily attributed to the lowered body temperatures during exercise.

SAA is a fast-responding acute-phase protein that starts to rise 6 h after an inflammatory stimulus and peaks after 36–48 h. SAA can be used to assess health and acute (sub)clinical disease in horses [32,33]. All the horses in the present study had plasma SAA concentrations below the reference range (<20 mg/L) on Day 1. None of the horses had an increase in SAA concentration after the first day of training in the heat, nor on Day 14. This indicates that the acclimatisation protocol used in this study did not elicit a relevant inflammatory response.

A limitation of the study is that only four horses participated in it. However, the two other studies evaluating the responses of horses before and after acclimation used four to six horses as well. As these studies are logistically and practically challenging, it was not practical to include more horses. In addition, the RH in the present study could not reach comparable levels to the hot and humid conditions presented by the other two studies (around 80–85% RH) [15,17]. This was due to the fact that the indoor riding arenas consisted of sand footing, and the moisture from the humidifiers was partly absorbed by the footing. For that reason, the RH in the present study was around 50–60%.

As the current study was not performed on a treadmill in an equipped laboratory, some parameters could not be measured at all or continuously. The sweat rate and the onset of the sweat rate were therefore not possible to measure. Nevertheless, to ascertain the applicability of laboratory-based acclimation studies in real-world scenarios, conducting the current study is of paramount importance. As fitness, age, and breed have significant effects on the physiological responses to heat [11,12], it is necessary to investigate whether Warmblood sport horses respond in a similar manner to heat acclimation. In addition, the authors of this study chose elite horses for several reasons: (1) high-level competition horses are transported across the globe to participate at international events, challenged with sudden large climate changes to cope with, (2) Warmblood sport horses may respond differently compared to research horses, (3) these specific Warmblood horses were highly fit for competition and may respond differently than less fit or lower-level horses, and (4) it is not likely that high-level competition horses will prepare for thermally challenging climates by using treadmill exercise due to the fact that they are not used to this type of exercise, and that in itself may increase the risk of injuries when suddenly induced for 1 h a day for 14 days. In addition, even when accustomed to treadmill exercise, this will influence their original training programme, limiting other training types or training times, which will affect their performance. All horses involved in this study exhibited exemplary performance during the subsequent Olympic Games, successfully completing their respective events while attaining qualifications and placements. Finally, when using heated indoor arenas, riders are also exposed and exercising for one hour a day for 14 days in the hot and humid environment, which may lead to their acclimation as well.

For all these reasons, it was important to assess whether acclimation in a heated indoor arena with submaximal exercise following the normal training routine of the horses would elicit the physiological adaptations necessary to prepare for thermally challenging climates.

## 5. Conclusions

The present study is the first to show that an acclimation period of 14 days for 60 min a day in a heated indoor arena (32 ± 1 °C, 50–60% RH) for elite sport horses can be used during real-life preparation for a high-level competition to significantly reduce thermal strain. After the acclimation period, the four Olympic horses showed a decrease in heart rate, rectal temperatures, and LA concentrations, an increase in the time taken to reach peak rectal temperatures, and an increase in heat storage. In addition, monitoring closely how individual horses respond to thermally challenging situations is crucial, as not all horses respond in the same manner, and some horses might be more heat-intolerant. Furthermore, as only a limited number of horses participated in this study and in other acclimation studies, the sometimes large individual responses underscore the need for further investigation on this topic, as with climate change, more horses will face hot and humid conditions.

## Figures and Tables

**Figure 1 animals-14-00546-f001:**
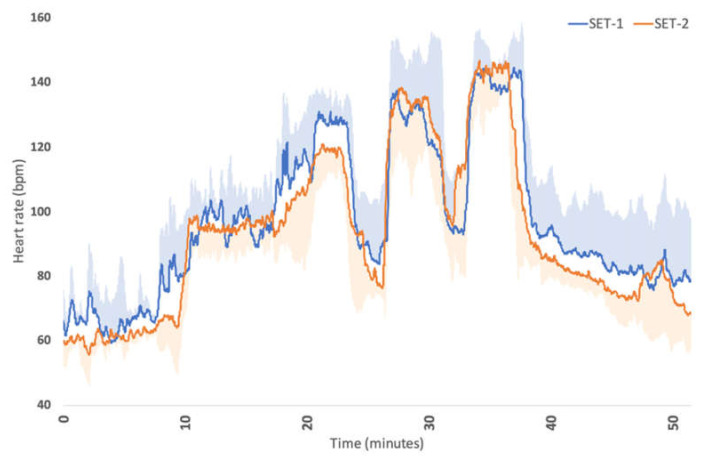
Average heart rate (bpm) and sd (shaded area) of the four Olympic horses before (SET 1; blue line) and after (SET 2; orange line) acclimation.

**Figure 2 animals-14-00546-f002:**
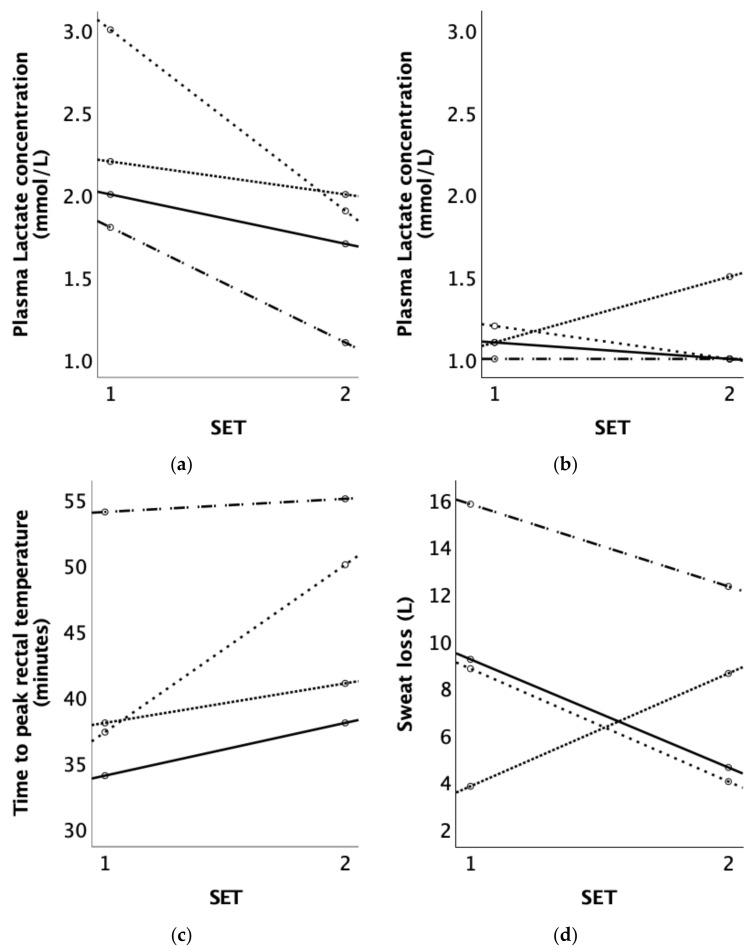
(**a**) Plasma lactate concentration (mmol/L) measured directly after the discipline specific phase and (**b**) plasma lactate concentration (mmol/L) measured after 10 min of recovery at a walk; (**c**) time to peak T_rec_ (minutes) and (**d**) sweat loss (litres) of the four Olympic horses before (SET 1) and after acclimation (SET 2). Each line represents an individual horse: Horse 1, Horse 2, Horse 3, and Horse 4.

**Figure 3 animals-14-00546-f003:**
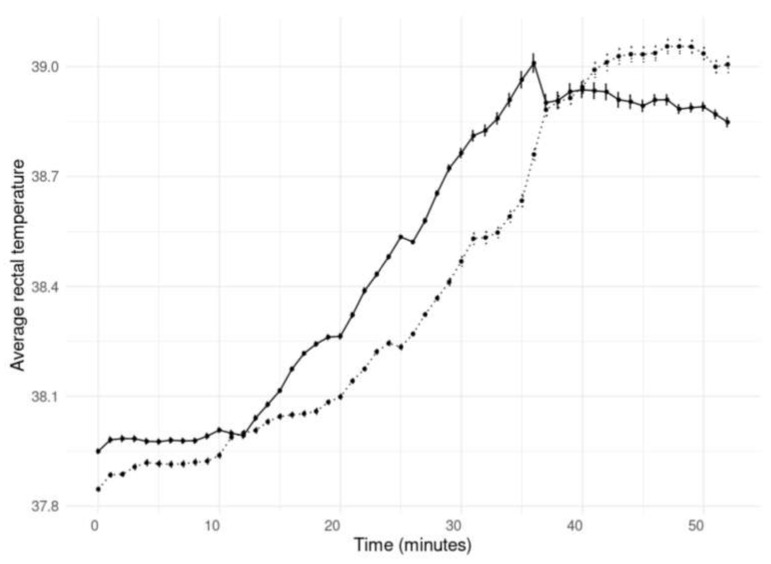
Average rectal temperatures and sd of the four Olympic horses before (SET 1; black line) and after (SET 2; dotted line) acclimation.

**Figure 4 animals-14-00546-f004:**
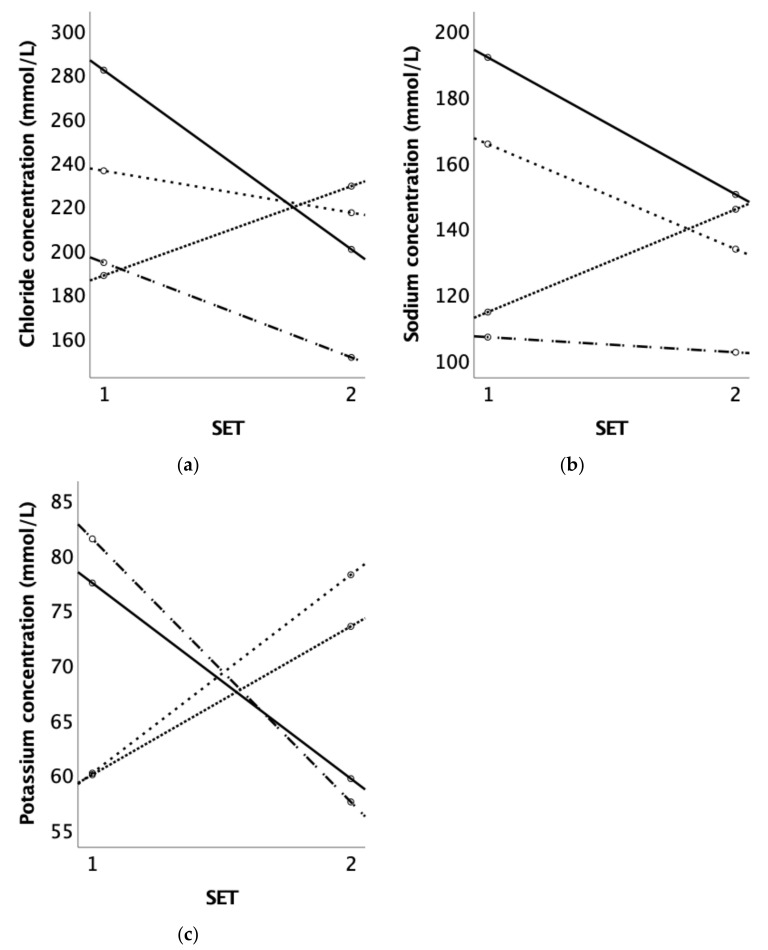
(**a**) Chloride concentration (mmol/L), (**b**) sodium concentration (mmol/L), and (**c**) potassium concentration (mmol/L) of the four Olympic horses before (SET 1) and after acclimation (SET 2). Each line represents an individual horse: Horse 1, Horse 2, Horse 3, and Horse 4.

**Table 1 animals-14-00546-t001:** Mean and standard deviation of total heat storage (S_tot_) at the end of the most intense exercise bout (discipline-specific phase; S_ex_) and after 10 min of recovery (S_rec_) for the four Olympic horses before (SET 1) and after acclimation (SET 2).

Acclimation	S_tot_ (kJ/m^2^)	S_ex_ (kJ/m^2^)	S_rec_ (kJ/m^2^)
SET 1	379.5 ± 96.2	491.4 ± 237.7	−4.3 ± 29.0
SET 2	549.4 ± 135.0	539.7 ± 102.1	9.7 ± 72.5

**Table 2 animals-14-00546-t002:** Mean and standard deviation of the sweat electrolyte concentrations on the gluteus skin of the four Olympic horses before acclimation on Day 1 (SET 1), Day 2, and after acclimation on Day 14 (SET 2).

Acclimation Day	Sodium (mmol/L)	Chloride (mmol/L)	Potassium (mmol/L)
Day 1 (SET 1)	152.5 ± 40.8	225.0 ± 43.4	69.7 ± 11.3
Day 2			
Day 14 (SET 2)	140.8 ± 21.6	199.2 ± 34.3	67.2 ± 10.2

## Data Availability

Individual data are unavailable due to privacy or ethical restrictions, as a limited number of horses were used in the study, and the horses participated in the Olympic Games and can be identified based on the data; this is not desirable due to privacy concerns and regulations related to these Olympic horses.

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
