# Peer review of "Effect of a 14-Day Period of Heat Acclimation on Horses Using Heated Indoor Arenas in Preparation for Tokyo Olympic Games"

_animals, 2024, doi:10.3390/ani14040546_

Round 1
Reviewer 1 Report
Comments and Suggestions for Authors
Thank you for an interesting and very practical study that provides meaningful outcomes for horse people. I have only had concerns with the English, and so have made many editorial corrections on the marked pdf, and have also a few queries that need to be addressed with revisions to wording.

see above
Reviewer 2 Report
Comments and Suggestions for Authors
Well written paper, but some weaknesses:
Low number of horses.
RH not high enough to reflect real conditions anticipated.
No control animals, perhaps 4 similar horses should have been exercised in the indoor arena without heat or humidity.
Was the training surface prior to the study exactly the same as the indoor arena?, same density and depth?
Was there any radiant heat from the heaters? / similar to direct sunlight.
